# Polyhydroxyalkanoates (PHAs) from Endophytic Bacterial Strains as Potential Biocontrol Agents against Postharvest Diseases of Apples

**DOI:** 10.3390/polym15092184

**Published:** 2023-05-04

**Authors:** Lyudmila Ignatova, Yelena Brazhnikova, Anel Omirbekova, Aizhamal Usmanova

**Affiliations:** Faculty of Biology and Biotechnology, Al-Farabi Kazakh National University, Almaty 050038, Kazakhstan; lyudmila.ignatova@kaznu.edu.kz (L.I.);

**Keywords:** eco-friendly bioplastic, physiochemical properties, antifungal activity, apple blue mold

## Abstract

Due to the increasing use and accumulation of petrochemical plastics in the environment and the rapid depletion of natural resources, microbial polyhydroxyalkanoates have great potential to replace them. This study provides new insights in the field of obtaining of polyhydroxyalkanoates (PHAs) from endophytic bacterial strains and applying them as potential biocontrol agents against postharvest diseases of apples. Two strains—*Pseudomonas flavescens* D5 and *Bacillus aerophilus* A2—accumulated PHAs in amounts ranging from 2.77 to 5.9 g L^−1^. The potential to use low-cost substrates such as beet molasses and soapstock for PHA accumulation was shown. The PHAs produced by the *Ps. flavescens* D5 strain had pronounced antagonistic activity against *Penicillium expansum* (antifungal property = 62.98–73.08%). The use of PHAs as biocontrol agents significantly reduced the severity of apple blue mold, especially in the preventive treatment option.

## 1. Introduction

In recent years, the increasing interest in environmental sustainability and green technologies has caused the need to develop innovative materials and technologies. Polyhydroxyalkanoates (PHAs) are structurally varied biodegradable polyester compounds—the only bioplastics obtained with the use of microorganisms [1,2]. Various Gram-positive and Gram-negative bacteria, fungi, and microalgae are able to synthesize and accumulate PHAs as intracellular granules using different substrates. Sugars are metabolized through a number of different metabolic pathways, which produce PHA precursors. Fatty acid substrates are metabolized through *β*-oxidation and de novo synthesis of fatty acids into PHAs [3].

Depending on their chain length, PHAs are divided into short-chain-length (scl) with 3–5 carbon atoms, medium-chain-length (mcl) with 6–14 carbons, and long-chain-length (lcl) with more than 14 carbons. In comparison with scl-PHAs, which are crystalline and have typical thermoplastic properties, mcl- and lcl-PHAs have lower molecular mass, low crystallinity and melting points, and are regarded as elastomers [2,3]. The PHAs consist of more than 150 monomers that can be combined to form homopolymers, statistical copolymers, and block copolymers, which provide endless variations of structures, along with unique physicochemical properties and functions of the biopolymers. Such a variety of structures and properties of the PHAs specifies a high potential for use in various areas of medicine, the food industry, and agrobiotechnology [1,2].

Compared to chemical plastics, PHAs have a number of advantages, such as being non-toxic, biodegradable, biocompatible, and environmentally friendly materials of the next generation. However, the production cost of PHAs is more expensive than the production of chemical plastics. The pursuit and development of effective producers of bacterial strains, the selection of cheaper raw materials, varying cultivation conditions, and the use of various advanced energy-efficient technologies for the isolation and purification of PHAs can reduce the cost of PHA production [1,3]. There are data about the use of various renewable resources as substrates, including organic, agricultural, dairy, and fruit wastes, as well as sewage, vegetable oil, animal fats, used cooking oil, etc. [1,4].

Annually, the losses of agricultural products in the world from fungal diseases during storage and transportation are estimated at several billion USD. In most countries, there is an unfavorable trend towards their growth; therefore, technological approaches that reduce the harmful effects of fungal diseases are under investigation [5,6,7]. Even though apple tree diseases during the growing season are successfully controlled by various methods of plant protection, there are no effective techniques to limit specific diseases during fruit storage. Annually observed storage rots manifest aggressiveness; they are characterized by high adaptive abilities—including to fungicides—and have low specificity in relation to the host. One of the most common pathogens is *Penicillium expansum*, which causes blue mold [6,7,8]. *P. expansum* produces an array of mycotoxins that have adverse effects on human health, including acute (convulsions, pulmonary congestion, edema, gastrointestinal bleeding, etc.) and chronic (e.g., genotoxic, neurotoxic, immunosuppressive, carcinogenic, and teratogenic) symptoms [6].

Effective microorganisms and their metabolites are used as biological methods to protect apple fruits from rot caused by *P. expansum* [6,8]. At the same time, despite the intensive study of PHAs, the issue of their biological activity remains insufficiently studied. There are data about the antibacterial properties of PHAs, while their antifungal activity has not been sufficiently studied [9,10].

The present study focuses on the investigation of PHAs’ properties and the possibilities of their application as biocontrol agents against postharvest diseases of apples. The main objectives of the present study are as follows: (1) Selection of low-cost substrates for effective PHA synthesis by endophytic bacterial strains. (2) Characterization of the physicochemical properties of PHAs. (3) The study of antifungal activity of PHAs against *P. expansum*.

This research presents novel insights in the field of the production and application of microbial polymers, deepening and expanding the existing knowledge. It was shown that more expensive carbon sources could be replaced with low-cost substrates such as beet molasses and soapstock, which can significantly reduce the cost of fermentation processes. An innovative eco-friendly method of protecting apples from postharvest infections by using PHAs is proposed. It was found that the obtained PHAs were effective against apple blue mold in both in vitro and in vivo experiments.

## 2. Materials and Methods

### 2.1. Materials

The objects of the study were endophytic bacterial strains isolated from the leaves of peppermint (*Méntha piperíta*)—*Bacillus aerophilus* A2 (accession number OQ569360)—and the flowers of common chicory (*Cichórium intybus*)—*Pseudomonas flavescens* D5 (accession number OP642636) [11].

All reagents used for the synthesis and characterization of PHAs were obtained from Sigma-Aldrich Company Ltd. (Dorset, UK).

### 2.2. Methods

#### 2.2.1. PHA Production Assay

The strains were grown in mineral media containing different carbon sources—such as glucose, olive oil, beet molasses, and soapstock—at 10 g L^−1^. The composition of the medium (g L^−1^) was as follows: 0.5 (NH_4_)_2_SO_4_; 0.4 MgSO_4_*7H_2_O; 9.65 Na_2_HPO_4_·12H_2_O; 2.65 KH_2_PO_4_. Additionally, 1 mL of a trace element solution of the following composition (g L^−1^) was added to the medium: 0.05 MnCl_2_·4H_2_O, 20.0 FeCl_3_·6H_2_O, 10.0 CaCl_2_·H_2_O, 0.03 CuSO_4_·5H_2_O, and 0.1 ZnSO_4_·7H_2_O in 0.5 N HCl. The strains were incubated for 48 h at 30 °C and 180 rpm, under aerobic conditions [12].

Chloroform and sodium hypochlorite were used to extract PHAs from bacterial cells. The mixture was incubated at 30 °C and 180 rpm for 90 min, and then centrifuged at 5000× *g* for 18 min. PHAs were precipitated by a double volume of isopropyl alcohol and dried [13].

The PHA production was quantified by weighing the precipitate and expressed in g per L. Bacterial cell pellets were dried to estimate the dry cell weight (DCW), which was expressed in g per L. The percentage of PHA content was estimated as the percentage composition of PHAs present in the DCW, according to the following formula [14]:$$\mathrm{PHA}\;\mathrm{content}\;\left(\%\right)=\frac{\mathrm{Dry}\;\mathrm{weight}\;\mathrm{of}\;\mathrm{PHAs}\;\left(g/L\right)}{\mathrm{Dry}\;\mathrm{Cell}\;\mathrm{Weight}\;\left(g/L\right)\;}\times 100\%$$

#### 2.2.2. Fourier-Transform Infrared (FTIR) Characterization of PHAs

Inverse Fourier-transform infrared spectroscopy was performed using a Carry 660 spectrophotometer (Agilent, Santa Clara, CA, USA) in the wavelength range from 800 to 4000 cm^−1^, equipped with a Germanium crystal (Ge ATR crystal); the number of reference scans through the air was 8, and the sample was scanned 24 times.

#### 2.2.3. Thermogravimetric Analysis of PHAs

Thermogravimetric analysis (TGA) was performed using Labsys EVO thermogravimetric equipment (Setaram, Caluire-et-Cuire, France) with a temperature range from 30 to 600 °C and a heating rate of 10 °C min^−1^ in a nitrogen atmosphere (N_2_ flow rate = 40 mL min^−1^) [15].

#### 2.2.4. In Vitro Antifungal Activity Tests

The antifungal properties of polyhydroxyalkanoates against *P. expansum* were tested using the shake-flask method [10].

First, 0.15 g of polyhydroxyalkanoate was diluted with 15 mL of phosphate buffer solution pH 7.4. A 1 mL suspension of *P. expansum* (10^5^ spores mL^−1^) was added to the solution. After 24 h, samples with a volume of 100 µL were taken and distributed into Petri dishes with nutrient agar. The samples were incubated at 32 °C for 24 h. Antifungal activity was calculated using the following formula:Antifungal Property = (1 − CFU_PHA_/CFU_control_) × 100

The antifungal activity of the PHA powders was estimated by the plate-hole diffusion method described by Balouiri, with slight modifications [16]. Briefly, 100 μL of fungal spore suspension containing 10^5^ spores/mL^−1^ of *P. expansum* was inoculated on agar plates. Then, 100 μL of PHA suspension at a concentration of 500 µg mL^−1^ was added to each previously made hole. The size of the inhibition zone against the phytopathogen was measured.

#### 2.2.5. In Vivo Antagonistic Assay

The pathogen *P. expansum* was grown on a PDA medium for 7 days at 25 °C, and then 10 mL of sterilized distilled water was added to each Petri dish. The mycelium was carefully scraped off. The concentration of the resulting conidial suspension was adjusted to 2 × 10^8^ spores/mL^−1^ using a hemocytometer. An aqueous solution of PHA was prepared at a concentration of 500 μg/mL.

The apple fruits were divided into 4 groups; five repeated fruits were used for each treatment, and the analysis was repeated twice. The first group of fruits was inoculated only with the pathogen. The second group was inoculated with PHA 24 h before infection with the pathogen, as a preventive treatment. The third group of fruits was inoculated with the pathogen and PHA simultaneously. The fourth group was inoculated with PHA 24 h after infection with the pathogen, as a therapeutic treatment [17].

The apple fruits’ surfaces were sterilized, and then holes with a diameter of 5 mm and a depth of 5 mm were made using a sterilized cork drill. Next, 100 µL of conidial suspension of *P. expansum* (2 × 10^8^ spores/mL^−1^) and 100 µL of PHA suspension (500 µg/mL) were inserted into the holes. Then, all fruits were incubated in humid conditions at 25 °C for 10 days. The effectiveness of the PHA treatment was evaluated on the 5th and 10th days of apple incubation according to the following indicators: weight loss and disease severity (DS).

For weight loss determination, all fruits were labeled and weighed on day 0, and then the same labeled fruits were weighed on days 5 and 10. The results were expressed as a percentage of the difference in weight loss between day 0 and day 5 or 10 [18].

To estimate disease severity (DS) the apple fruits were classified by a five-point scale as described by Safari [19]. The following formula was used for DS calculation:$$\mathrm{DS}\;\left(\%\right)=\frac{\sum \;\left(\mathrm{Group}\;\mathrm{score}\times \mathrm{Number}\;\mathrm{of}\;\mathrm{damaged}\;\mathrm{fruits}\;\mathrm{in}\;\mathrm{each}\;\mathrm{group}\right)}{\mathrm{Total}\;\mathrm{number}\;\mathrm{of}\;\mathrm{evaluated}\;\mathrm{fruits}\;\times \;\mathrm{Highest}\;\mathrm{DS}\;\mathrm{score}\;}\times 100\%$$

#### 2.2.6. Statistical Analysis

Statistical analysis was performed using the Statistica software v. 10.0 (TIBCO Software Inc., Palo Alto, CA, USA). All data are presented as the mean ± standard deviation. Data analysis was performed using single-factor analysis of variance (ANOVA), followed by the use of Tukey’s HSD test for multiple comparisons.

## 3. Results

### 3.1. Selection of Low-Cost Substrate for PHA Synthesis by Endophytic Bacterial Strains

PHAs are a diverse group of eco-friendly polyesters synthesized by a variety of microorganisms. However, their high production costs compared with plastics derived from petrochemicals make the extensive production and commercial application of PHAs challenging [2,19]. Because PHAs are growth-associated products, optimization of culture media conditions is considered to be essential for efficient microbial growth and PHA accumulation. The substrate used in the medium plays a key role in PHA biosynthesis [14,19]. In the present study, two bacterial endophytic strains were tested for their ability to produce PHAs when growing in media with different carbon sources, including commercial substrate (glucose), vegetable oil (olive oil), byproducts of the sugar industry (beet molasses), and byproducts of the oilseed processing industry (soapstock). The results of the chemical composition of beet molasses indicated that the total sugar content was 78.42%, including saccharose (41.34%), glucose (21.96%), and fructose (4.04%). The composition of soapstock was as follows: total fat 92.44%, non-fat impurities 0.9%.

The investigation of the influence of various carbon sources on bacterial growth revealed that the *Ps. flavescens* D5 strain produced the highest biomass when olive oil was used as the sole source of carbon, followed by soapstock, whereas glucose had the least biomass yield. The strain *B. aerophilus* A2 showed the highest DCW on the medium with soapstock, followed by molasses (Table 1).

The PHA contents varied widely, from 37.6 to 76%, depending on the substrate and producers used (Table 1). The *Ps. flavescens* D5 and *B. aerophilus* A2 strains accumulated PHA in amounts ranging from 2.77 to 5.9 g L^−1^, which is significantly higher than the values mentioned in previous studies for other strains (0.1–5.1 g L^−1^) [20,21].

The highest values of PHA production for both strains were achieved when soapstock was used as the substrate (Table 1). Soapstock has a multicomponent composition, rich in various fatty acids. Microorganisms synthesize PHA using several metabolic pathways, depending on the substrate, including sugar metabolism, β-oxidation of fatty acids, and de novo synthesis of fatty acids [2,3]. The rich composition of the soapstock makes it possible to fully realize the biosynthetic potential of the studied bacterial strains. Similar to our results, previous studies showed effective production of PHAs on media containing different agro-industrial wastes [14,21,22,23].

It is known that substrates entail major costs in PHA production [2,19]. The obtained results allow us to conclude that more expensive carbon sources can be replaced with low-cost substrates such as beet molasses and soapstock. Thus, the usage of agro-industrial wastes and byproducts as carbon sources during the biosynthesis of PHAs by bacterial strains can significantly reduce the cost of fermentation processes.

### 3.2. Characterization of Physicochemical Properties of PHAs

#### 3.2.1. FTIR Characterization of PHAs

The chemical composition of the extracted powdered PHA samples was analyzed by using FTIR spectrum data. There have been some studies that used FTIR spectroscopy to characterize PHAs [24,25,26,27,28], but only a few studies indicating the influence of the composition of the nutrient media—specially supplied with different carbon sources—on the characteristics of PHAs [29,30].

In our work, PHA samples extracted from *B. aerophilus* A2 and *Ps. flavescens* D5 were analyzed. They were incubated on nutrient media supplemented with glucose, beet molasses, and soapstock byproducts as carbon sources. The FTIR analysis, interpretation, and comparison of the obtained spectral data with previous studies and spectral databases allowed us to identify several peaks that have been described in many studies as characteristic of PHAs (Figure 1 and Figure 2).

In all PHA samples, most of the peaks presented in the wavenumber range from 1000 to 3000 cm^−1^ are known as specific regions of PHAs.

The peaks present in the ester, methylene, and terminal hydroxyl groups typically indicate the polymeric structure of PHAs [31]. The absorption peaks at 1721, 2920, and 3268 cm^−1^ are the characteristic peaks of carbonyl (C=O), methine (-CH), and hydroxyl (-OH) groups, respectively. In detail, the following absorption bands were found for all powder samples: bands at 3248 cm^−1^ and 3274 cm^−1^, which correspond to stretching of a terminal OH group; bands at 2932 cm^−1^ and 2951 cm^−1^, which indicate C-H stretching in methyl and methylene groups; a band at 1650 cm^−1^, corresponding to C=O stretching of an ester group; and a C-O band at 1280 cm^−1^. The band at 1648 cm^−1^ defines the stretching of the amide group at the ester carbonyl (C=O) position, and the absorption band at 848 cm^−1^ could correspond to the β-glycosidic linkage between the sugar monomers [31]. The broad peak centered around 3250 cm^−1^ denotes hydrogen-bonded O–H strands. It is known that a carboxylic acid produces characteristic broad 0-H absorption as well as intense carbonyl stretching absorption. Because of the unusually strong hydrogen bonding in carboxylic acids, the broad 0-H stretching frequency is shifted to about 3000 cm^−1^. This wide absorption of 0-H gives a characteristic of inflated shape to peaks in the C-H stretching region.

Only the FTIR spectrum of the PHA sample extracted from *B. aerophilus* A2 incubated on soapstock as a carbon source showed various other peaks. The characteristic band at 2818–2951 cm^−1^ shows the presence of antisymmetric and symmetric stretching of –CH bonds (alkanes) in the –CH_3_ and –CH_2_ groups.

There are several studies indicating that a carbon source is one of the cultivation conditions that considerably influences the composition and properties of PHAs. Nair et al., investigated the characterization of PHAs produced by *Bacillus subtilis* with the addition of glucose, mannitol, and sugarcane molasses as carbon sources [32]. The FTIR analysis indicated that the same polymer was produced from the different carbon sources, although intensity of the peaks differed a little, which can be explained by a highly ordered crystalline or more amorphous structure.

Javers and Karunanithy obtained PHA samples produced by *Pseudomonas putida* KT217 on a condensed corn-solubles-based medium with glycerol water or sunflower soapstock [33]. In particular, the addition of sunflower soapstock changed the monomeric composition of the polymer—specifically, 3-hydroxyoctanoate monomer became dominant and made the composition of the PHA less saturated.

#### 3.2.2. Thermal Properties

The thermal properties of the PHAs produced by *B. aerophilus* A2 and *Ps. flavescens* D5 were analyzed using a differential thermal analyzer (DTA) by TGA. TGA was used to analyze the thermal stability of the extracted PHAs.

Recent studies have shown that one of the main factors for the successful fermentation of bacteria is a well-constructed production medium, which significantly affects the mechanical and thermal properties of PHA materials [34,35].

To describe the polymers’ thermal stability, the temperatures at 5% and 10% weight loss—denoted as T_d(5%)_ and T_d(10%)_, respectively—and the temperature at minimum decomposition rate (T_dmin_) were used. The results of the TGA experiments are reported in Table 2.

From Table 2, it can be seen that degradation temperature due to the 10% weight loss of PHA samples ranged from 100.5 to 174.6 °C. The minimum degradation temperature range for all three substrates was 32.1–103.0 °C. The comparison of the results of weight loss for PHAs from *B. aerophilus* A2 showed that the PHA sample derived from soapstock substrate had higher thermal stability due to the higher T_10%_ temperature. Similar results were seen for *Ps. flavescens* D5 grown on soapstock substrate. Moreover, the weight loss at 600 °C was higher among other samples. This may be related to the peculiarity of the FTIR spectrum of that PHA sample, which showed various additional peaks indicating antisymmetric and symmetric stretching bonds in the –CH_3_ and –CH_2_ groups.

The results of the PHA sample from *Bacillus aerophilus* A2 with beet molasses substrate showed lower thermal stability due to the low T_5%_ and T_10%_ temperatures, as well as lower weight loss at 600 °C than the other samples. Thus, it was established that at 100 °C there is a loss of the moisture mass of the sample; then, with the increase in temperature, a gradual loss of the mass is observed.

### 3.3. Antagonistic Activity of PHAs In Vitro against Penicillium expansum

Natural polymers commonly do not have antibacterial properties, with the exception of chitosan, whose antibacterial effect due to the positively charged amino groups in its structure has been established [36]. Therefore, natural polymers—especially PHAs—are most often studied as polymer matrices for the inclusion of antimicrobial agents [37], while their own antimicrobial activity—and especially their antifungal activity—is insufficiently studied.

The results of the study of the antagonistic activity of PHA samples produced by the *Ps. flavescens* D5 and *B. aerophilus* A2 strains showed that they act differently. As shown in Table 3 and Figure 3, the PHA produced by the *B. aerophilus* A2 strain manifested slight antifungal activity against the *P. expansum* test culture. The PHA produced by the *Ps. flavescens* D5 strain displayed pronounced bioactivity. The results of microbiological studies of PHAs on solid nutrient media are consistent with the results of the evaluation of antifungal activity in a phosphate buffer followed by plating on Sabouraud agar. The *Ps. flavescens* D5 PHA sample showed activity against *P. expansum*, causing the formation of inhibition zones, as shown in Figure 3 and in Table 3. The results of the shake-flask method demonstrated that the use of PHA ensures the loss of the phytopathogen by 62.98–73.08% and does not significantly depend on the composition of the culture medium of the strain-producer.

Not all PHAs possess antimicrobial or antiviral activity. The possible mechanisms of biocontrol activity can be related to the destruction of the microbial wall/membrane and the leakage of intracellular contents and changes in transmembrane potential [38,39]. The antimicrobial properties of PHAs are believed to be related to their monomers—hydroxyl-substituted fatty acids, which act as anionic surfactants. They are absorbed by microbial cells, disrupt their permeability, and inhibit their growth and development. In addition, the presence of antimicrobial activity of PHAs is associated with the length of the carbon chain. More often, antimicrobial activity is associated with scl-PHAs [37].

In the present study, only PHAs produced by the *Ps. flavescens* D5 strain showed pronounced antifungal activity, so they were used for further experiments in vivo.

### 3.4. Evaluation of the Possibility of Using PHAs As Biocontrol Agents against Postharvest Diseases of Apples

Over the past two decades, various effective methods of postharvest control of apple blue mold caused by *P. expansum* have been described. Most of these works are concerned with microorganisms and their metabolites that exert a biocontrol function against phytopathogens [6,8]. To test the possibility of using PHAs produced by the *Ps. flavescens* D5 strain as biocontrol agents, three treatments of apple fruits were proposed: combined inoculation of PHA and the phytopathogen *P. expansum*, preventive treatment of PHA 24 h before inoculation of the phytopathogen, and therapeutic treatment with PHA 24 h after inoculation of the phytopathogen [8].

Inoculation of apple fruits with spores of *P. expansum* resulted in the development of blue mold lesions, which appeared as small soft spots of light brown color. Then, the spots were covered with a fungal white mycelium, and eventually covered with grayish-green conidia. The affected pulp had a typical mildew smell and a watery structure spreading deep into the fruit. Control fruits inoculated with sterile water showed no signs of disease (Figure 4).

In this study, it was shown that the obtained PHAs were effective against apple blue mold in experiments in vivo, resulting in a decrease in disease severity and/or weight loss. The results obtained with the inoculation of *P. expansum* and PHA in apple fruits are shown in Table 4 and Figure 4. The severity of infection was determined by weighing the treated apple fruits and determining the area of the affected areas. In all cases, *P. expansum* treatment caused the development of rot symptoms on inoculated fruits. However, the severity of symptoms decreased under simultaneous inoculation of fruits with phytopathogens and PHA.

The use of a biocontrol agent 24 h before the inoculation of fruits with *P. expansum* (preventive treatment) revealed that PHA caused a noticeable twofold decrease in disease on the 10th day compared to the control fruits. When fruits were inoculated with *P. expansum* simultaneously with the bioagent, the DS was 44%. Finally, when using PHA as a therapeutic treatment (24 h after infection with the pathogen), the lowest inhibitory activity of the polymer (DS = 48%) was shown.

In all treatments, a decrease in the diameter of the lesions was observed, especially with preventive treatment compared to the control treatment with only the pathogen. Moreover, during combined inoculation of *P. expansum* with PHA into the wound of the apple, there was a decrease in the mass of rotting tissues around the wound, as well as in the depth of their spread into the fruit (Table 4, Figure 4).

Weight loss and DS are important indicators to evaluate the effectiveness of the application of biocontrol agents. After infection with the pathogen *P. expansum* and the development of the disease, the weight loss of treated and control apple fruits was detected. However, the weight loss of the PHA-treated fruits was significantly lower than that of the controls, indicating a healing process initiated by the application of the PHA. This was probably due to the physical barrier effectively inhibiting water transpiration and promoting wound healing [40].

Our results allow us to propose a new microbial PHA for practical use as a potential biocontrol agent against postharvest diseases, and also as a basis for the synthesis of new polymer materials with highly effective antifungal activity. This innovative method has significant advantages over other methods, including the use of live microorganisms. This method is simple and easy to use, whereas certain conditions to preserve their viability and biological activity should be ensured for the use of microorganisms.

## 4. Conclusions

It was found that two bacterial endophytic strains were able to produce PHAs when growing in media with different carbon sources, including commercial substrate (glucose), vegetable oil (olive oil), byproducts of the sugar industry (beet molasses), and byproducts of the oilseed processing industry (soapstock). The results of FTIR analysis showed that in all PHA samples most of the peaks were present in the wavenumber range of 1000 to 3000 cm^−1^. The thermal properties of the PHAs were studied by thermogravimetric analysis, and it was established that at 100 °C there is a loss of the moisture mass of the sample and, then, with an increase in temperature, a gradual loss of the mass is observed. A new field of application of PHAs as biocontrol agents to protect apples from postharvest infections was demonstrated.

## Figures and Tables

**Figure 1 polymers-15-02184-f001:**
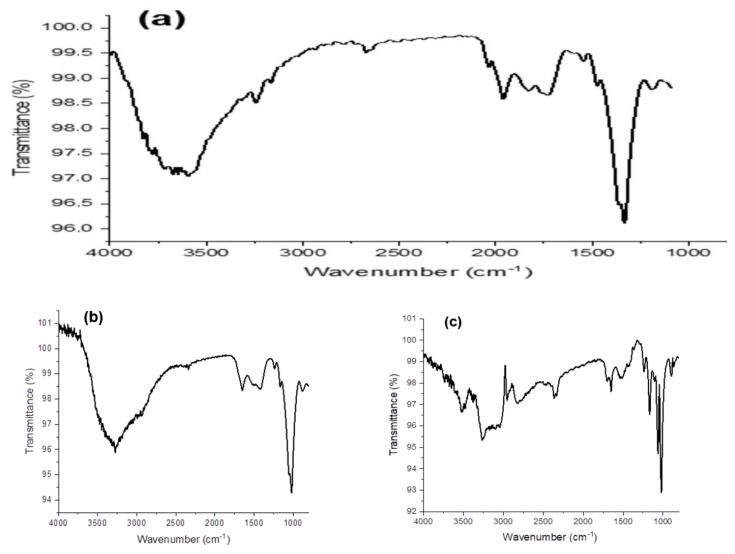
FTIR spectra of PHA samples extracted from *Bacillus aerophilus* A2 incubated on various substrates: (**a**) glucose; (**b**) beet molasses; (**c**) soapstock.

**Figure 2 polymers-15-02184-f002:**
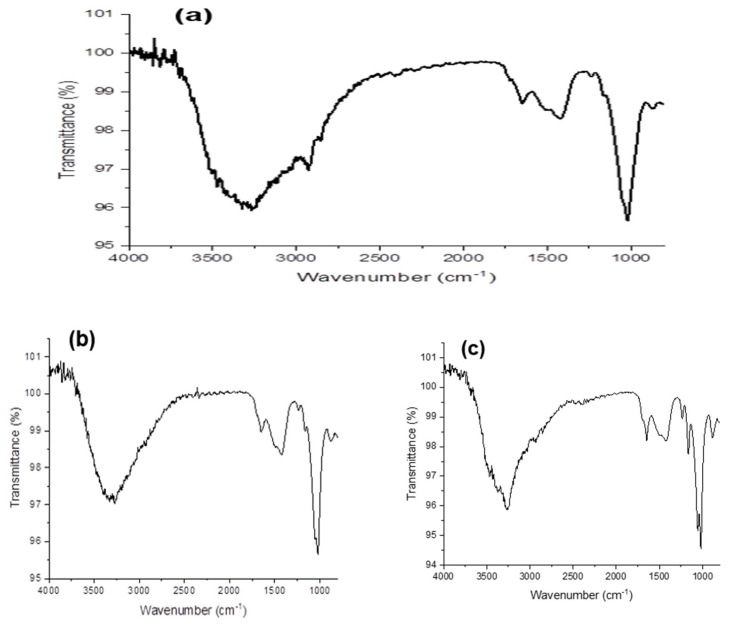
FTIR spectra of PHA samples extracted from *Pseudomonas flavescens* D5 incubated on various substrates: (**a**) glucose; (**b**) beet molasses; (**c**) soapstock.

**Figure 3 polymers-15-02184-f003:**
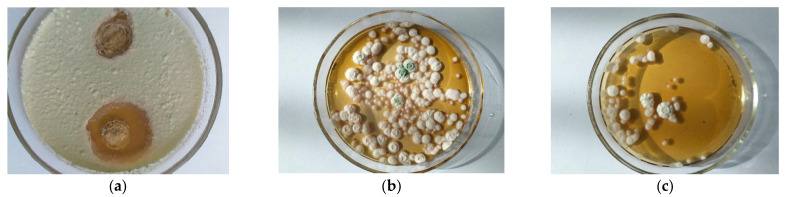
Antifungal activity of PHA obtained from *Pseudomonas flavescens* D5 strain against *Penicillium expansum* (**a**) using the plate-hole diffusion method and the shake-flask method; (**b**) control plate without PHA; (**c**) with PHA.

**Figure 4 polymers-15-02184-f004:**
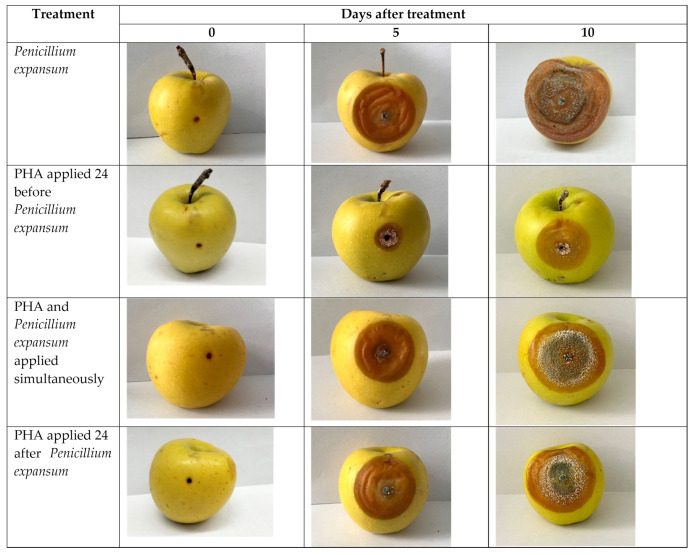
In vivo antifungal efficiency of the PHA bioagents against *Penicillium expansum* on apple fruits.

**Table 1 polymers-15-02184-t001:** Effects of different carbon sources on bacterial growth and PHA production.

Substrate	*Pseudomonas flavescens* D5			*Bacillus aerophilus* A2		
	DCW, g L^−1^	PHA Content, %	PHA Production, g L^−1^	DCW, g L^−1^	PHA Content, %	PHA Production, g L^−1^
Glucose	3.83 ± 0.19 a	72.2 ± 3.2 c	2.77 ± 0.1 a	6.2 ± 0.25 a	73.2 ± 3.1 c	4.54 ± 0.2 b
Olive oil	10.2 ± 0.4 d	37.6 ± 1.5 a	3.83 ± 0.15 b	6.13 ± 0.2 a	54.3 ± 2.4 a	3.33 ± 0.15 a
Beet molasses	7.23 ± 0.2 b	60.8 ± 2.2 b	4.4 ± 0.18 c	6.17 ± 0.18 a	75.1 ± 3.5 c	4.63 ± 0.14 b
Soapstock	8.2 ± 0.3 c	72 ± 2.5 c	5.9 ± 0.15 d	8.3 ± 0.34 b	61 ± 3.2 b	5.07 ± 0.22 c

Values are given as the mean ± standard deviation. Values followed by the same letter(s) are not significantly different at *p* ≤ 0.05 according to Tukey’s HSD test.

**Table 2 polymers-15-02184-t002:** Thermal properties of PHA samples.

Substrate	Weight Loss at Final Degradation T 600 °C, mg/%	Td (5%), °C	Td (10%), °C	Tdmin, °C
*Bacillus aerophilus* A2				
Glucose	−5.8/59	118.1	155	82.7
Beet molasses	−4.64/38.2	74.0	100. 5	32.1
Soapstock	−5.40/40.9	160.5	173.5	103.0
*Pseudomonas flavescens* D5				
Glucose	−4.51/55.0	114.2	147.0	79.9
Beet molasses	−6.67/55.4	146.0	165.4	77.3
Soapstock	−8.49/60.2	155.0	174.6	85.5

**Table 3 polymers-15-02184-t003:** Antifungal tests for PHA powder.

Carbon Source	Antifungal Property, %		Zone of Inhibition, (cm)	
	*Ps. flavescens* D5	*B. aerophilus* A2	*Ps. flavescens* D5	*B. aerophilus* A2
Glucose	69.27 ± 3 b	-	1.06 ± 0.02 a	0.36 ± 0.01 d
Olive oil	65.12 ± 2.5 a	-	1.25 ± 0.05 b	0.22 ± 0.01 a
Beet molasses	62.98 ± 2 a	-	1.11 ± 0.01 a	0.31 ± 0.01 c
Soapstock	73.08 ± 3.2 b	-	1.18 ± 0.02 b	0.28 ± 0.01 b

Values are given as the mean ± standard deviation. Values followed by the same letter(s) are not significantly different at *p* ≤ 0.05 according to Tukey’s HSD test.

**Table 4 polymers-15-02184-t004:** Effect of inoculation of the apple fruits in vivo with the PHA and *Penicillium expansum*.

Treatment	Weight Loss, %		Disease Severity, %	
	5 Days	10 Days	5 Days	10 Days
*Penicillium expansum*	1.92 ± 0.07 d	4.18 ± 0.2 d	36 ± 4 c	64 ± 6 c
PHA applied 24 before *Penicillium expansum*	0.95 ± 0.03 a	2.13 ± 0.08 a	20 ± 2 a	32 ± 2 a
PHA and *Penicillium expansum* applied simultaneously	1.31 ± 0.04 b	3.08 ± 0.1 b	28 ± 2 b	44 ± 4 b
PHA applied 24 after *Penicillium* *expansum*	1.52 ± 0.05 c	3.45 ± 0.12 c	32 ± 2 c	48 ± 4 b

Values are given as the mean ± standard deviation. Values followed by the same letter(s) are not significantly different at *p* ≤ 0.05 according to Tukey’s HSD test.

## Data Availability

The data that support the findings of this study are available upon request from the corresponding author.

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
