# Peer review of "Polyhydroxyalkanoates (PHAs) from Endophytic Bacterial Strains as Potential Biocontrol Agents against Postharvest Diseases of Apples"

_polymers, 2023, doi:10.3390/polym15092184_

Round 1
Reviewer 1 Report
After careful reviewing of this manuscript, I have come to conclusion that herein the authors proposed a new and cost-friendly methodology to save apples from post-harvesting diseases by using Polyhydroxyalkanoates (PHAs) derived from endophytic bacterial strains. Detailed characterization of PHAs has also been included. Overall, the manuscript is interesting and demands publication. However, there are few issues, which should be addressed before its possible publication. Therefore, I would like to suggest major revision based on the following comments:
1. In abstract, the information regarding FTIR and TGA can be avoided. Rather, these information are more suitable for Conclusions.
2. In introduction, a brief discussion should be included regarding blue mold, i.e., its adverse effects on human beings if consumed.
3. The authors claimed easy and cost-friendly synthesis of PHA from bacterial stains. However, the method described in stains formation is time-consuming and requires a lost of chemicals. Therefore, can this described process be at all scaled up?
4. In Table 1, what do the alphabets a,b,c… mean?
5. Fig. 1 and 2 have been directly copied and pasted from instrument. This is very poor way of representation. I would like to suggest the authors to collect datasheet and plot it in Origin or other graphing software. Also, it’s more convenient to read FTIR through %Transmittance. Therefore, please convert the Absorbance vs wavenumber plot to %Transmittance vs. wavenumber plot.
6. The broad peak centred around 3250 cm–1 denotes hydrogen bonded O–H str. The concept of hydrogen bonding should be discussed.
7. 1H NMR of PHAs should be studied to elucidate their structures.
8. MALDI-TOFF data will be very essential to find out molecular weight distribution of polymeric PHAs.
Author Response
Dear Reviewer,
We are very grateful for consideration of our manuscript entitled: “Polyhydroxyalkanoates (PHAs) from endophytic bacterial strains as the potential biocontrol agents against postharvest diseases of Apple”.
We thank very much for your careful and thorough review. We have revised our manuscript accordingly and provided clarifications and additional data. Appropriated changes have been introduced to the manuscript.
We hope that you will find our responses satisfactory.
Sincerely yours,
Authors

Reviewer 2 Report
The study discusses the potential use of microbial polyhydroxyalkanoates (PHAs) as a substitute for petrochemical plastics, due to their ability to accumulate in large amounts and their potential as biocontrol agents against postharvest diseases of apples. The study focuses on two endophytic bacterial strains, Pseudomonas flavescens D5 and Bacillus aerophilus A2, which were able to accumulate PHA from low-cost substrates such as beet molasses and soapstock. The PHAs produced by Ps. flavescens D5 strain showed antifungal properties and significantly reduced the severity of apple blue mold in preventive treatment. The study also includes analyses of the FTIR and thermal properties of the PHAs.
In my point of view the paper is good for publication in the current format since the novelty of work can open a new field of study for the interested scientists.
Author Response
Dear Reviewer,
We are very grateful for consideration of our manuscript entitled: “Polyhydroxyalkanoates (PHAs) from endophytic bacterial strains as the potential biocontrol agents against postharvest diseases of Apple”. We thank very much for your careful and thorough review.
Sincerely yours,
Authors
Reviewer 3 Report
The work is interesting and presents another application of PHAs as an antifungal. To improve the quality and comprehensibility of the manuscript some changes should be made. Please find enclosed comments and suggestions.

Author Response

(The authors gave the same response as above.)

Round 2
Reviewer 1 Report
The authors worked hard to satify most of my comments. Therefore, now this manuscript is suitable for publication.